# Detection of Pathogenic Leptospires in Water and Soil in Areas Endemic to Leptospirosis in Nicaragua

**DOI:** 10.3390/tropicalmed5030149

**Published:** 2020-09-18

**Authors:** Byron Flores, Karla Escobar, José Luis Muzquiz, Jessica Sheleby-Elías, Brenda Mora, Edipcia Roque, Dayana Torres, Álvaro Chávez, William Jirón

**Affiliations:** 1Centro Veterinario de Diagnóstico e Investigación (CEVEDI), Departamento de Veterinaria y Zootecnia, Escuela de Ciencias Agrarias y Veterinarias, Universidad Nacional Autónoma de Nicaragua-León (UNAN-León), Carretera a La Ceiba 1 Km al Este, León 21000, Nicaragua; karlamassielez93@gmail.com (K.E.); jessicasheleby@gmail.com (J.S.-E.); brenda.mora@ev.unanleon.edu.ni (B.M.); cibt0509@gmail.com (D.T.); alchasi_91@hotmail.com (Á.C.); williamjiron@gmail.com (W.J.); 2Department of Animal Pathology, Faculty of Veterinary Sciences, Universidad de Zaragoza, Miguel Servet 177, 50013 Zaragoza, Spain; muzquiz@unizar.es; 3Research Centre on Health, Work and Environment (CISTA), National Autonomous University of Nicaragua, León 21000 (UNAN-León), Nicaragua; edipciaroque@yahoo.es; 4Animal Welfare Department, School of Agricultural Sciences, Catholic University of the Dry Tropic (UCATSE), Estelí 31000, Nicaragua

**Keywords:** *Leptospira* spp., leptospirosis, water, soil, river, Nicaragua

## Abstract

In Nicaragua, there are ideal environmental conditions for leptospirosis. The objective of this investigation was to detect pathogenic and saprophytic leptospires in water and soil samples from leptospirosis-endemic areas in Nicaragua. Seventy-eight water and 42 soil samples were collected from houses and rivers close to confirmed human cases. *Leptospira* spp was isolated in Ellinghausen–McCullough–Johnson–Harris (EMJH) culture medium with 5-fluororacil and positive samples were analyzed through PCR for the *LipL32* gene, specific for pathogenic leptospires (P1 clade). There were 73 positive cultures from 120 samples, however only six of these (5% of all collected samples) were confirmed to be pathogenic, based on the presence of the *LipL32* gene (P1 clade). Of these six pathogenic isolates, four were from Leon and two from Chinandega. Four pathogenic isolates were obtained from water and two from soil. This study proved the contamination of water and soil with pathogenic leptospires, which represents a potential risk for public health.

## 1. Introduction

Leptospirosis is a zoonosis that significantly affects public health worldwide. The number of cases in recent years has increased, accounting for more than 500,000 each year [1]. This infectious disease is usually endemic in regions with tropical and subtropical climates due to the high humidity [2]. Climatic variation, rural environment and work occupation are considered important risk factors [3,4].

This disease is classified as an occupational risk for veterinarians, sewer cleaners, miners, animal breeders and soldiers. However, recreational aquatic activities have also been identified as a risk factor as a result of immersion in contaminated water [5].

Animals carrying leptospires eliminate the bacteria through urine and contaminate the environment. This is how humans can become infected through direct or indirect contact with contaminated water or soil [6,7,8]. Nevertheless, some studies have had difficulty isolating pathogenic leptospires from the environment, as was the case of a 2019 study conducted in Japan, where only one pathogenic *Leptospira* (subclade P1) was isolated from 100 water samples [9]; while in a study from Indonesia, one out of 73 samples (soil and water) was confirmed as an intermediate pathogenic *Leptospira* (subclade P2) [10]. Nicaragua has had outbreaks of leptospirosis since 1995; from 2007 to 2011, 568 leptospirosis alerts were registered in the world, of which 53 were located in Nicaragua [11,12]. The annual incidence of this disease in the country is 23.3 per million, although only some cases were confirmed due to limited surveillance and misdiagnosis [13].

Severe climatological events such as floods and hurricanes that have taken place in Nicaragua in recent decades are considerably related to the presentation of clinical cases in humans, especially in the departments of León and Chinandega (western part of the country) [14]. In addition, it is recognized as a common cause of acute feverish illness in the country [12]. The aim of this investigation was to detect pathogenic leptospires in the water and soil from endemic areas in Nicaragua. The identification of sources of infection in the environment is critical for establishing preventive measures to prevent the appearance of new cases in animals and humans.

## 2. Materials and Methods

A cross-sectional study was carried out in the departments of León and Chinandega (in western Nicaragua), classified as a critical area for the occurrence of leptospirosis outbreaks [15], and Jinotega (in the north), due to the high number of confirmed human cases in 2017. The sampling was carried out from July 2017 to February 2018. Seventy-eight water samples and 42 soil samples were collected, within a radius of 100 m from homes with a case of human leptospirosis reported by the Ministry of Health (MINSA), and in rivers that are used for recreational purposes. In the specific case of the “Telica river” in the department of León, the samples were taken from four different points in the river with high recreational concurrence.

### 2.1. Samples Collection

Regarding water, 29 samples were taken directly from storage containers inside the houses, 11 from wells and 3 from puddles. Thirty-five river samples were taken in shaded places unaffected by direct solar rays. One liter of water was collected in sterile bottles and transported at 4 °C.

Concerning soils, 20 samples of wet soils were taken from rain puddles and 22 soil samples were taken from areas of the rivers under abundant shade. These samples were transported at room temperature in 50 mL conical tubes.

### 2.2. Samples Analysis

All the samples were analyzed in the leptospirosis laboratory in the Centro Veterinario de Diagnóstico e Investigación (CEVEDI), Universidad Nacional Autónoma de Nicaragua, León (UNAN-León).

The water samples were filtered with 0.22 µm membranes and then 2 or 3 drops (approximately 50 µL) were inoculated in 5 mL of Ellinghausen–McCullough–Johnson–Harris (EMJH) liquid medium (Difco^®^, Detroit, MI, USA) combined with 40 µg/mL sulfamethoxazole, 20 µg/mL trimethoprim, 5 µg/mL amphotericin B, 400 µg/mL phosphomycin, and 100 µg/mL 5-fluorouracil [16]. They were incubated at a temperature of 28–30 °C and reviewed weekly with a darkfield microscope. Subcultures were performed, through passes, and a sample was considered positive when growth was obtained in at least one of the passes and discarded as negative after 16 weeks without growth.

For isolation from soil samples, 3 g of each soil sample which was properly identified, was weighed and placed in 15 mL sterile conical tubes to which 10 mL of sterile distilled water was added, mixed until completely dissolved and held upright for one hour. Subsequently, the supernatant was taken, and the same procedure was performed as the isolation from water samples.

### 2.3. Polymerase Chain Reaction (PCR)

The samples that were positive to the isolation were used to carry out the DNA extraction. Leptospires isolated with 7 days of incubation were centrifuged at 17,500× *g* for 10 min; the supernatant was discarded and 200 µL of the precipitate was taken to carry out the DNA extraction following the indications of the commercial kit UltranClean^®^ Blood Spin (MO BIO, Carlsbad, CA, USA). The result was stored at −20 °C until analysis.

As for the amplification, a previously described method was used [17] with the primers *LipL32*-270F (5′-CGCTGAAATGGGAGTTCGTATGATT-3′) and *LipL32*-692R (5′-CCAACAGATGCAACGAAAGATCCTT-3′), which flank a 423 bp product to the *LipL32* gene, present only in pathogenic species (P1 clade). The reaction mixture was made in a volume of 25 µL, adding 12.5 µL of PCR-Master Mix 2x (Promega, Madison, WI, USA), 5.5 µL nuclease-free water, 1 µL *LipL32*-270F primers, 1 µL *LipL32*-692R primers and 5 µL DNA. The conditions for amplification were: an initial denaturation of 94 °C for 5 min, 40 cycles of 95 °C for 30 s, 50 °C for 30 s and 72 °C in 1 min; culminating with an elongation of 72 °C in 7 min. To read the product, 2% agarose gel electrophoresis stained with ethidium bromide was prepared.

### 2.4. Statistical Analysis

Relative frequencies were calculated with their respective Confidence Interval (95% CI) and the contingency tables were used for cross-categorical variables. In addition, significance was calculated using Fisher’s exact test to compare the isolation and PCR results between departments, type of sample (water vs. soil) and source (river, stored water, well and puddle). The data were stored and analyzed in SPSS version 21.

## 3. Results

We obtained 73 *Leptospira*-positive cultures out of 120 environmental samples (60.83%, 95% CI = 51.68–69.98). A higher frequency of positives was found in the water samples with 67.94% (53/78) compared to the soil samples with 47.6% (20/42), (*p* = 0.036). The highest frequency of positive isolates was found in the department of León with 70.3% (26/37), while the lowest percentage of positive samples was obtained in Jinotega 43.2% (16/37), (*p* = 0.033). The results by source of the samples (stored water, puddle, well, river), did not show significant differences (*p* ≥ 0.05), with 50.90% (29/57) positive river samples and 65.20% (15/23) positivity in puddle samples. A higher frequency of positive samples (*p* = 0.032) was found in the month of October with 74.40% (32/43) compared to November with 30.0% (6/20). The analysis for each type of sample (water or soil) showed that 20/29 samples of stored water and 21/35 samples of water taken from rivers were positive to the isolation of spirochetes (*p* ≥ 0.05), while 12/20 soil samples taken from puddles and 8/22 taken from rivers showed growth (*p* ≥ 0.05).

Regarding the presence of pathogenic leptospires (as tested by PCR *LipL32*), 6/120 (5.00%, 95% CI = 0.68–9.31) of the total samples were positive, 2/46 samples were positive in Chinandega, 0/37 in Jinotega, and 4/37 in León, (Table 1). Pathogenic species were identified in 2/42 soil samples and 4/78 water samples. In rivers, 2/35 positive water samples were found (Table 2), specifically in Telica river, which is located in the department of León (Figure 1). Detailed information about each sample in this study is shown in Table A1.

## 4. Discussion

The presence of pathogenic leptospires has been evidenced in different water sources and soil types and identified as a potential risk factor in rural and urban areas around the world. In this study, environmental samples from rivers and peri-urban areas of the departments of León, Chinandega and Jinotega in Nicaragua were analyzed to determine the presence of pathogenic and saprophytic leptospires.

Conventional PCR targeting the *LipL32* gene specific for pathogenic *Leptospira* (subclade P1) [18], identified 5% (6/120) of *LipL32*-positive *Leptospira*. Similar studies have also been able to identify *Leptospira* in the environment, such as that of Saito et al. in the Philippines in 2013, who detected pathogenic leptospires in 11 out of 23 wet soil samples from coastal areas after a storm [19]. Other studies also found low frequencies in the isolation of pathogenic (P1) and intermediate pathogenic species (P2) in water and soil samples [9,10,20]. The difficulty in isolating leptospires P1 could be attributed to a probably weaker adaptation to the environmental conditions of pathogenic species compared to saprophytic species (S2, S3), [20]. In addition, isolation from environmental samples is challenging due to the slow growth of leptospires and the overgrowth of co-existing microorganisms [16].

The detection of the *LipL32* gene is useful for the epidemiological surveillance of pathogenic species in environmental samples, where the majority of *Leptospira* isolates are saprophytic species [21]. However, other researchers found that the pathogens cluster is heterogeneous, being composed of both virulent and low-virulence strains, which is why in vivo models of infection are necessary for a better characterization of isolates [22].

All samples positive for pathogenic leptospires were isolated in León and Chinandega (none in Jinotega). Flores et al., found similar results, revealing that León and Chinandega exhibited high percentages of positivity for pathogenic leptospires isolation in the samples of domestic animals and rodents during the period 2007–2013 [23]. Other studies similarly showed that the western zone exhibited the highest number of cases of human leptospirosis in the country [15]. These data suggest that in the western region there are ideal factors for the *Leptospira* cycle to be effectively carried out, since reservoirs are present to ensure their maintenance in the environment. For this reason, it is pertinent to implement measures that identify infected animals and contaminated water and soil sources that represent risk areas and exposure of the population to pathogenic leptospires.

One sample from stored water (inside the houses) was PCR positive. These results could be associated with those of rural areas in Nicaragua, where the population stores water for daily use in open barrels, allowing potential contamination with rodent’s urine [24]. A study carried out in settlements in Guatemala City found that housewives obtained the highest percentage of anti-*Leptospira* antibodies, suggesting that the presence of animals in domestic environments represents a risk factor through the contamination of water sources or soils, thus directly to the bacteria [25] exposing the population.

Moreover, one of the well water samples was *Lipl32*-positive. This contamination could have occurred due to the presence of infected bats that urinate inside the well [26]. This would constitute a great risk for public health, since wells are in most cases the only source of water used for animal and human consumption.

As for soil analysis, the two *Lipl32*-positive samples corresponded to a wet soil obtained from rain puddles, both taken from a backyard with evidence of rearing pigs, a common cultural practice in rural areas of Nicaragua. Swine, like other domestic animals, excretes *Leptospira* through urine, contaminating soils, thus making human infection possible [27]. Furthermore, Schneider et al. found an association between the predominant volcanic soil type in western Nicaragua with a high incidence of human Leptospirosis. They hypothesized that it may be due to Leptospira’s ability to survive longer in neutral to alkaline soils [15].

Although previous studies describe that in the rainy months (April–November) human leptospirosis increases [15], in this study, no association was found between the months and the *Lipl32* PCR results: the positive river samples were collected during the dry season (January) while the positive samples from stored water and rain puddles were collected during the rainy season (April and October respectively). These results suggest that pathogenic leptospires represent a greater risk for human infection in the peri-domicile environment in the rainy season, while in the dry season, rivers and other sources of water for recreational use should be monitored annually.

In this study, samples from rivers used for recreational activities and near cases of human leptospirosis were analyzed. In water from these rivers, two samples positive to pathogenic leptospires were identified, more precisely, in “Telica” river in León. Both samples were taken from places where cattle come to drink water. As cattle usually urinate where they drink, this could explain the presence of pathogenic *Leptospira* in the water [23]. In this same department in 2007, there was an outbreak of human leptospirosis associated with the presence of pathogenic leptospires in the “La Leona” river, a place that in the dry season is also used by the surrounding population for recreational activities [11]. These findings confirm that aquatic recreational activities in this region represent an important risk factor for the population. Therefore, the monitoring of these rivers must be carried out regularly in locations linked to human activity for the prevention and control of new outbreaks of leptospirosis, as evidenced by a 2016 study in which pathogenic leptospires were isolated from a river in Thailand by sampling at various points along the river route [28]. Similarly, in this study, the “Telica” river was sampled at four points along its route (Figure 1). In both studies, it was possible to isolate pathogenic leptospires in specific points along the river route, demonstrating good diagnostic sensitivity by allowing different perspectives on the area.

## 5. Conclusions

A low frequency of pathogenic *Leptospira* (P1) was found in the environmental samples from western and central Nicaragua, even though the sampling occurred in human leptospirosis endemic areas. This could be linked to the difficulty of pathogenic species’ isolation from environmental samples. However, the presence of pathogenic *Leptospira* detected in river samples used for recreational purposes poses a risk to public health.

## Figures and Tables

**Figure 1 tropicalmed-05-00149-f001:**
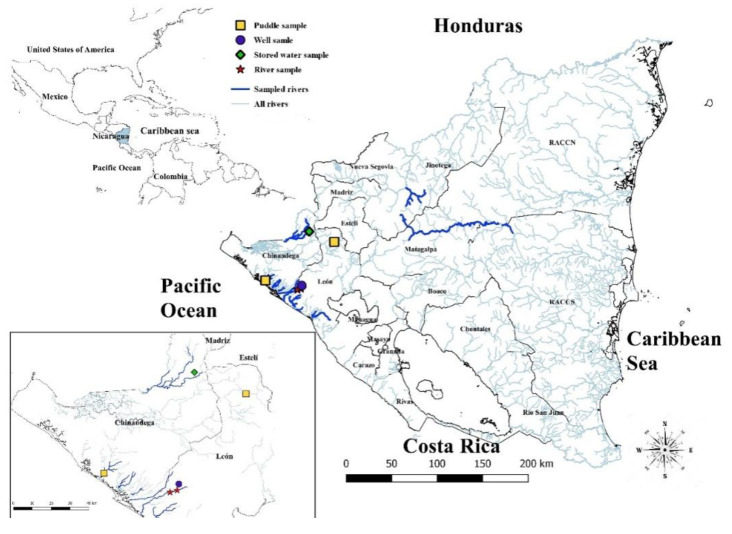
Georeferencing of points where pathogenic leptospires species were obtained by source of the samples.

**Table 1 tropicalmed-05-00149-t001:** Results of *Leptospira* spp. isolation and pathogenic *Leptospira* detection (*LipL32* PCR) in environmental samples from Nicaragua (*n* = 120).

Variables	Values of Each Variable	Isolation		*p* Value for Positive Isolation Comparison *	PCR Positive	*p* Value for Positive PCR Comparison *
		Negative	Positive			
**Departments**	Chinandega	15	31	0.036	2	0.254
	Jinotega	21	16		0	
	León	11	26		4	
**Type of sample**	Water	25	53	0.033	4	0.528
	Soil	22	20		2	
**Source of the sample**	Stored water	9	20	0.162	1	0.772
	Puddles	8	15		2	
	Wells	2	9		1	
	Rivers	28	29		2	
**Months**	January	5	10	0.032	2	0.256
	February	1	3		1	
	March	7	10		0	
	April	9	12		1	
	October	11	32		2	
	November	14	6		0	

*: From Fisher’s exact test.

**Table 2 tropicalmed-05-00149-t002:** Isolate and PCR *LipL32* result in teh water and soil samples by source collection.

Type of Sample	Source of the Sample	Isolation		*p* Value *	PCR Positive	*p* Value *
		Negative	Positive			
**Water**	Stored water	9	20	0.429	1	0.864
	Puddles	0	3		0	
	Wells	2	9		1	
	Rivers	14	21		2	
	Total	25	53		4	
**Soil**	Puddles	8	12	0.216	2	0.495
	Rivers	14	8		0	
	Total	22	20		2	

*: From Fisher’s exact test. PCR: polymerase chain reaction.

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
