# Peer review of "Detection of Pathogenic Leptospires in Water and Soil in Areas Endemic to Leptospirosis in Nicaragua"

_tropicalmed, 2020, doi:10.3390/tropicalmed5030149_

Round 1
Reviewer 1 Report
The authors of this study proposed to assess the frequency of identifying pathogenic leptospires from various soil or water samples from different districts in Nicaragua. The materials and methods is pretty straightforward and I don't have any significant concerns here. However, the authors completely lost me in the Results section. There are only 6 isolates that are confirmed to be pathogenic leptospires via PCR, yet the authors cite the isolation rate via culture as indicating a high percentage of positives. Isolating the leptospires is step 1 of 2 in identifying pathogenic leptospires. It appears that the authors have identified isolated a large number of saprophytic (nonpathogenic) leptospires, not pathogenic leptospires. Unless the authors are going to identify (by MLST, VNTR, or pulse-field gel electrophoresis) the specific serovars isolated in each culture and demonstrate that the LipL32 PCR failed to accurately identify these isolates, all they can state in this paper is that they isolated 6 pathogenic leptospires (a 5% isolation rate).
A couple of specific comments (however, grammar needs extensive editing):
- What is the purpose of limiting the sampling to 100 m from the domicile of an individual with leptospirosis? That assumes the person does not travel anywhere. It would be better to interview the person with leptospirosis and then sample the sites most likely linked to exposure.
- The paragraph/sentences beginning around line 115 report data that is not shown (or contradicted) by Table 1.
- In line 126, I believe the authors intended to write that 0/16 were positive for the gene, not negative.
Author Response
Please see the attachment
Dear editor of Tropical medicine and infectious disease and dear reviewer 1, below we provide a cover letter to explain, point by point, the details of the revisions in the manuscript, Detection of pathogenic leptospires in water and soil in areas endemic to leptospirosis in Nicaragua. We are grateful for your support.
Best regards
Reviewer 1
The authors of this study proposed to assess the frequency of identifying pathogenic leptospires from various soil or water samples from different districts in Nicaragua. The materials and methods is pretty straightforward and I don't have any significant concerns here. However, the authors completely lost me in the Results section. There are only 6 isolates that are confirmed to be pathogenic leptospires via PCR, yet the authors cite the isolation rate via culture as indicating a high percentage of positives. Isolating the leptospires is step 1 of 2 in identifying pathogenic leptospires. It appears that the authors have identified isolated a large number of saprophytic (nonpathogenic) leptospires, not pathogenic leptospires. Unless the authors are going to identify (by MLST, VNTR, or pulse-field gel electrophoresis) the specific serovars isolated in each culture and demonstrate that the LipL32 PCR failed to accurately identify these isolates, all they can state in this paper is that they isolated 6 pathogenic leptospires (a 5% isolation rate).
In abstract we changed to “In PCR, 6/120 of total samples were LipL32 positive (6/73 positive isolation, 8.21%, 95% CI: 1.23 - 15.20)”
In results we changed to “In the PCR, 6/120 (5.00%, 95% CI = 0.68-9.31) of total samples were LipL32 positive”
We modify the conclusions, it is highlighted in blue color
Other changes related to this point are highlighted in blue in the document
A couple of specific comments (however, grammar needs extensive editing):
Grammar was revised
- What is the purpose of limiting the sampling to 100 m from the domicile of an individual with leptospirosis? That assumes the person does not travel anywhere. It would be better to interview the person with leptospirosis and then sample the sites most likely linked to exposure.
Thanks for your observation.
The main reason for taking samples in a diameter of 100 around the case of human leptospirosis was to delimit a sampling area, although it is true that we cannot assume that people did not travel to another place, most of their time, they coexist in the peridomestic setting, this was suggested by the National Zoonosis Committee. The criterion was used as an approximation.
- The paragraph/sentences beginning around line 115 report data that is not shown (or contradicted) by Table 1.
These data are reported in Table 2 (Table 2 was modified)
- In line 126, I believe the authors intended to write that 0/16 were positive for the gene, not negative.
Was changed to 0/16 isolates were positive for the gene

Reviewer 2 Report
Flores and Coll conducted a field and laboratory study to assess the presence of leptospires in 120 environmental samples and to estimate the frequency of pathogenic leptospires. They evidenced the presence of pathogenic leptospires in water and soil in endemic areas, emphasizing that contaminated environment represents a potential risk to public health.
While this work constitutes a valuable contribution to a better knowledge of leptospirosis epidemiology in Nicaragua several inconsistencies needs to be addressed in order to make the manuscript more intelligible.
Comment 1 (major):
I do not understand the logic behind the procedure used by the authors to isolate the leptospires from the water and soil samples.
“Water (and soils) samples were filtered using 0.22 um membranes and incubated in EMJH supplemented with STAFF”.
After filtration through 0.22um, almost all the contaminant microbiota and most of the lesptospira are retained, (less than 1 % of leptospira are able to pass 0.22um filter). Therefore, the STAFF treatment after 0.22um filtration does not seem relevant (which in addition imposes a negative pressure on leptospira growth). Is there any particular reason why the authors adopted such a procedure?
Filtration through 0.45um would have been more relevant: more Leptospira could have gone through the filter, (therefore significantly increasing the overall positivity rate which is surely impacted by the 0.22ul filtration) and in this case the use of STAFF is appropriated since more contaminant bacteria are present.
I strongly recommend that the authors adopt procedures that have already been published and that appear to be more effective than the one used in this manuscript.
I can cite as an example the work done by Thibeaux and Coll, 2018 (DOI : 10.3389/fmicb.2018.00816 )
Comment 2 (major):
I noticed that DNA used as a template for the LipL-32 PCR is extracted from 200 ul of a culture supernatant. It is more than likely that the positives culture contain several differences species of leptospires (a mixture of species from P1-P2-S1-S2 clade).
Why the authors didn't isolate the bacteria on semi-solid EMJH prior performing their PCR on DNA from axenic cultures?
Comment 3 (major):
Recently, the discovery of new species of leptospira has made it possible to refine their phylogeny and in particular to highlight variations in their virulence (see PMID: 29310748 and PMID: 31120895). Thus, in Clade Pathogen P1, although all species present in this clade possess the LipL-32 gene, roughly half are virulent and are responsible for infections (in humans and mammals).
In their manuscript, the authors identify lipL-32 positive cultures as part of the clade pathogen, which is true, but have not studied their virulence (in animal models for example).
Thus, leads to a confusion between pathogen and virulence: the authors suggest that their isolates are virulent, which has not been demonstrated.
Either the authors provide evidences that their isolates are indeed capable of inducing leptospirosis disease in an animal model or they replace pathogenic isolates by LipL32 + isolates and clarify this point in the discussion.
Comment 4 (minor):
Lines 157-160:
In addition, the type of soil of volcanic origin in the studied areas and the humidity levels generated by rainfall in the rainy months provide highly favorable conditions for the conservation of Leptospira [13]. Studies such as that of Benecer et al., isolated leptospires in soils with 23.3%, a low percentage compared to this study, this is because the soils in their study were sandy and loam type which have little retention capacity of water [16].
There is no direct evidences that volcanic soils are more suitable for leptospira survival than sandy or loam type soils. It seems evident that soils play a major role in leptospira survival but so far, no major studies linking soil composition to leptospira survival have been published. I suggest the author to mitigate theses sentence.
Comment 4 (minor):
Lines 164-166: This sentence should be rephrase.
As I mentioned in comment 3, the presence of the LipL-32 genes does not prove that the isolate is virulent. A positive LipL-32 PCR just indicate that the studied isolate belong to Clade P1.
For example:
For this reason, conventional PCR targeting the lipL32 genes was performed to confirm that the isolates obtained belong the pathogen P1 clade, finding a percentage of positives of 8.2% (6/73).
Comment 4 (minor):
Lines 168-170.
To me, this sentence in unintelligible. Please rephrase. In addition, I would recommend the author to revise the English language thorough their manuscript.
Comment 5 (minor):
Line 34: A word is missing
“It is usually endemic” or “this infectious disease is usually endemic”…..
Line 88: kit UltranClean® Blood Spin de MO BIO, (USA), should be: kit UltranClean® Blood Spin from (MO BIO, USA),
Line 89: -20oC should be: -20°C
Line 124: 2/15 of puddle: these numbers seems false. Shouldn't it rather be:2/16
Line 125: in Chinandega 2/ isolates. The denominator is missing.
Round 2
Reviewer 1 Report
I appreciate that the authors made changes in the paper to reflect that only 5% of all samples contained pathogenic leptospires (as determined by the PCR). But, I feel that the authors are still too invested in their original manuscript (results and discussion) touting an isolation rate of 60.8%. If we accept that lacking the LipL32 gene classifies ~92% of the isolates as saprophytic, then any discussion of environmental contamination by cattle and rodents and risks associated with recreational activity on the water should not be discussed in context of a positive rate of isolating saprophytic leptospires. I'm actually more fascinated with a positive pathogenic leptospire identified in a stored water source--how was it stored, from where was it collected, how long was it sitting there, was it covered? And where were the "puddles"? That is poorly described. Yet soil samples from "puddles" had a 10% positive isolation rate for pathogenic leptospires. The authors state this could be related to farm animals, but were farm animals in contact with these puddles? Were these puddles temporary after a rainstorm or flood?
I accept that the authors may view identifying only 6 pathogenic leptospires as a disappointing result of their study and want to continue to focus on the higher isolation rate, but they do themselves a disservice by missing the relevance of these 6 pathogenic isolates. I would also be interested to know in which months these 6 pathogens were isolated. If they collected water samples from the same recreational water site 6 times over the course of their study and in one of those six samples (say, in February) was positive for a pathogenic leptospire, and a similar finding was reproducible with some of the other pathogenic leptospires, then that could be relevant information. If the pathogenic leptospires are scattered throughout the year, that's also relevant.
I still think it's appropriate to make the statement in the discussion about the LipL32 gene and the need for in vivo models. They could use that added sentence as the beginning of a paragraph (or two) discussing the high isolation rate of apparently saprophytic leptospires and introduce their concerns that these saprophytic leptospires may be a marker for pathogenic leptospires.
Reviewer 2 Report
Revisions carried out by the authors are satisfactory and have made the manuscript more intelligible.
Round 3
Reviewer 1 Report
This version is improved.
ABSTRACT
The abstract states it quite plainly that the purpose of this study is "to detect pathogenic leptospires" from different sources. Therefore, the authors still need a re-write. From Line 23-26, that all needs to be removed. If the authors want to replace this with, "There were 73 positive cultures from 120 samples, however only 6 of these (5% of all collected samples) were confirmed to be pathogenic based on the presence of the LipL32 gene. Of these six pathogenic isolates, 4 were from Leon and 2 from Chinandega. Four pathogenic isolates were obtained from water and two from soil. This study..."
I don't really understand the comparison of isolation of nonpathogenic leptospires from each district.
INTRODUCTION
I think the authors need to do a better job introducing the idea of isolating pathogenic leptospires from the environment (not from the wild animals) and including a couple of similar studies that may demonstrate the difficulties in isolating P1 clade from the environment. For instance, one 2019 study out of Japan isolated only 2 pathogenic leptospires from 50 water samples and neither of the isolates was known to have ever been isolated from human cases in Japan and in a 2019 study out of Jakarta they isolated only 1 intermediate pathogen out of 73 samples. This, ultimately, has to be the focus of this paper given the few positive samples.
MATERIALS AND METHODS
The authors establish the area of collection based on high occurrence of leptospirosis in these districts and even establish a set reason for where the collected--within 100 m of a home in which a case was identified and recreational waters. But, despite this criteria, they isolate only 6 pathogens.
Line 92: Celsius does not have the degree sign like Fahrenheit. It's just 20 C.
RESULTS
Line 117-119: To what does the p value <0.05 refer? What's being compared here?
And for Table 1. To what do the values refer? What's being compared?
This section should be organized better. If the authors want to start with the isolation results and then move to PCR results to show true pathogenic leptospires, that's how it should be done. But not a paragraph of isolation, then PCR, then isolation, then PCR.
DISCUSSION
The first paragraph is flawed here. The first sentence ends with "...to determine the presence of pathogenic leptospires", but then the authors open the second sentence to discuss the isolation of all leptospires, not just the pathogenic ones. There can't be a discussion about "critical and endemic" when presenting saprophytic leptospire data. The saprophytes have nothing to do with "a large number of cases of human leptospirosis".
Much of the second paragraph should just be incorporated into the first. Cut out any discussion of isolates here.
I think the Discussion needs to focus on the difficulty of environmental sampling and why that probably poorly documents the human risk as compared to looking at isolation from rodents, cattle, etc...
CONCLUSION
I don't agree with the conclusion. That's not what I concluded from this study. It's not a high frequency and there cannot be real conclusions drawn about seasonal risk.
Author Response
ABSTRACT
The abstract states it quite plainly that the purpose of this study is "to detect pathogenic leptospires" from different sources. Therefore, the authors still need a re-write. From Line 23-26, that all needs to be removed. If the authors want to replace this with, "There were 73 positive cultures from 120 samples, however only 6 of these (5% of all collected samples) were confirmed to be pathogenic based on the presence of the LipL32 gene. Of these six pathogenic isolates, 4 were from Leon and 2 from Chinandega. Four pathogenic isolates were obtained from water and two from soil. This study..."
I don't really understand the comparison of isolation of nonpathogenic leptospires from each district.
Thanks for the kindness to provide a specific suggestion, we have made the change
INTRODUCTION
I think the authors need to do a better job introducing the idea of isolating pathogenic leptospires from the environment (not from the wild animals) and including a couple of similar studies that may demonstrate the difficulties in isolating P1 clade from the environment. For instance, one 2019 study out of Japan isolated only 2 pathogenic leptospires from 50 water samples and neither of the isolates was known to have ever been isolated from human cases in Japan and in a 2019 study out of Jakarta they isolated only 1 intermediate pathogen out of 73 samples. This, ultimately, has to be the focus of this paper given the few positive samples.
We added
however, some studies have had difficulty isolating pathogenic leptospires from the environment, as is the case of a 2019 study out of Japan, where only 2 pathogenic leptospires from 100 water samples were isolated [9], while Widiyanti et al. (2019) isolated only 1 intermediate pathogenic (P2 clade) out of 73 water and soil samples in Indonesia [10].
MATERIALS AND METHODS
The authors establish the area of collection based on high occurrence of leptospirosis in these districts and even establish a set reason for where the collected--within 100 m of a home in which a case was identified and recreational waters. But, despite this criteria, they isolate only 6 pathogens.
We added in discussion
Other studies also found low frequencies in the isolation of pathogenic (P1) and intermediate pathogenic species (P2) in water and soil samples [9,10]. This discrepancy can be attributed to the fact that the analyzed samples in this study come from endemic areas, where a large number of human leptospirosis cases have been reported during the outbreaks in Nicaragua since 1995 [15,22]. On the other hand, the difficulty isolating leptospires P1 could be attributed to a probably weaker adaptation to environmental conditions of pathogenic species compared to saprophytic species (S2, S3), [20].
Line 92: Celsius does not have the degree sign like Fahrenheit. It's just 20 C.
Thanks, this change had already been made. is it possible that the reviewer is not seeing the updated version?
RESULTS
Line 117-119: To what does the p value <0.05 refer? What's being compared here?
Sorry we didn't find this in the place mentioned by the reviewer
And for Table 1. To what do the values refer? What's being compared?
Values are referring to the value for each analyzed variable and p values are for comparison of positive isolation and positive PCR frequency.
We added more detail in table 1
This section should be organized better. If the authors want to start with the isolation results and then move to PCR results to show true pathogenic leptospires, that's how it should be done. But not a paragraph of isolation, then PCR, then isolation, then PCR.
Thanks, the results were organized in such a way that the results of the isolation of Leptospira spp are described first and then the results of pathogenic leptospires.
DISCUSSION
The first paragraph is flawed here. The first sentence ends with "...to determine the presence of pathogenic leptospires", but then the authors open the second sentence to discuss the isolation of all leptospires, not just the pathogenic ones. There can't be a discussion about "critical and endemic" when presenting saprophytic leptospire data. The saprophytes have nothing to do with "a large number of cases of human leptospirosis".
Much of the second paragraph should just be incorporated into the first. Cut out any discussion of isolates here.
This was organized and improved
I think the Discussion needs to focus on the difficulty of environmental sampling and why that probably poorly documents the human risk as compared to looking at isolation from rodents, cattle, etc...
We added
On the other hand, the difficulty isolating leptospires P1 could be attributed to a probably weaker adaptation to environmental conditions of pathogenic species compared to saprophytic species (S2, S3), [20].
CONCLUSION
I don't agree with the conclusion. That's not what I concluded from this study. It's not a high frequency and there cannot be real conclusions drawn about seasonal risk.
We changed to
A low frequency of pathogenic leptospires (P1) was found, considering the origin of the samples from human leptospirosis endemic areas, confirming the difficulty of pathogenic species isolation from environmental samples, however, the presence of pathogenic leptospires in water samples used for recreational purposes pose a risk to public health.